# Risk and predictive factors for severe dengue infection: A systematic review and meta-analysis

**Kangzhuang Yuan, Yuan Chen, Meifeng Zhong, Yongping Lin\*, Lidong Liu**  **\***

Division of Clinical Laboratory Medicine, The First Affiliated Hospital of Guangzhou Medical University, Guangzhou, Guangdong, P.R. China

\* lin_y_p@hotmail.com (YL); lidong.liu@gzhmu.edu.cn (LL)

## Abstract

### Background

Dengue is a major public health issue worldwide and severe dengue (SD) is life threatening. It is critical to triage patients with dengue infection in the early stage. However, there is limited knowledge on early indicators of SD. The objective of this study is to identify risk factors for the prognosis of SD and try to find out some potential predictive factors for SD from dengue fever (DF) in the early of infection.

### Methods

The PubMed, Cochrane Library and Web of Science databases were searched for relevant studies from June 1999 to December 2020. The pooled odds ratio (OR) or standardized mean difference (SMD) with 95% confidence intervals (CI) of identified factors was calculated using a fixed or random effect model in the meta-analysis. Tests for heterogeneity, publication bias, subgroup analyses, meta-regression, and a sensitivity analysis were further performed.

### Findings

A total of 6,848 candidate articles were retrieved, 87 studies with 35,184 DF and 8,173 SD cases met the eligibility criteria. A total of 64 factors were identified, including population and virus characteristics, clinical symptoms and signs, laboratory biomarkers, cytokines, and chemokines; of these factors, 34 were found to be significantly different between DF and SD, while the other 30 factors were not significantly different between the two groups after pooling the data from the relevant studies. Additionally, 9 factors were positive associated with SD within 7 days after illness when the timing subgroup analysis were performed.

### Conclusions

Practical factors and biomarkers for the identification of SD were established, which will be helpful for a prompt diagnosis and early effective treatment for those at greatest risk. These

**Data Availability Statement:** All relevant data are within the paper and its Supporting Information files.

**Funding:** This work was funded by Guangzhou Science Technology and Innovation Committee (https://sop.gzsi.gov.cn/egrantweb/, NO. 201607010163 awards to Lidong Liu), Health and Family Planning Commission of Guangdong Province (http://wsjkw.gd.gov.cn/, NO. A2016448 awards to Lidong Liu) and Guangzhou Medical University (https://www.gzhmu.edu.cn/, NO.2014C24 awards to Lidong Liu). The funders had no role in study design, data collection and analysis, decision to publish, or preparation of the manuscript.

**Competing interests:** The authors have declared that no competing interests exist.

outcomes also enhance our knowledge of the clinical manifestations and pathogenesis of SD.

## Introduction

Dengue disease is a mosquito-borne viral infection caused by the dengue virus (DENV). Patients infected with DENV have a wide spectrum of clinical manifestations, ranging from asymptomatic to dengue fever (DF) or severe dengue (SD), including dengue hemorrhagic fever (DHF) and dengue shock syndrome (DSS) [1, 2]. The World Health Organization (WHO) estimated that approximately 2.5 billion people living in dengue-endemic countries [3]. With an increasing incidence of DENV infections each year, it was estimated that there were 390 million dengue infections per year, of which 96 million manifested symptomatically [4]; additionally, it was estimated that there were 565,900 disabilities and 9110 deaths in 2013 [5]. The first licensed recombinant, live-attenuated dengue vaccine (Dengvaxia) recently became clinically available. However, a high risk of adverse outcomes was found among vaccinated individuals who had not been previously exposed to dengue [6, 7]. Severe and fatal cases were consistently reported in some endemic areas, such as Southeast Asia, the Western Pacific, and the Americas [8–10]. It has been reported that the DSS mortality is 50 times higher than that of DF [11], and SD has been a leading cause of serious illness and death among children in some Asian and Latin American countries [3]. Previous data showed that the SD mortality would decrease from more than 20% to less than 1% if SD were identified and properly treated in a timely fashion [3]. Hence, the early prediction and recognition of severe cases are critical for dengue disease management.

To help clinicians evaluate the likelihood of severe disease, risk factors for SD have been reported, such as secondary infection, gastrointestinal pain, vomiting, diarrhea, intravascular leakage and bleeding [12]. Efforts have been consistently made to identify predictive markers for SD [13–15]. Although dengue with warning signs (WS) was referenced in the newly updated WHO guideline [1], a multicenter study reported that approximately 30% of adults with DF had WS and only 10% developed SD [16], while another study showed that the sensitivity and specificity of WS were 59–98% and 41–99%, respectively, when they were used to identify SD [17]. Numerous potential markers for SD have been reported but some have been inconclusive [18–20]. To distinguish SD from DF in the early of infection and try to find out some potential predictive factors, we conducted this systematic review and meta-analysis.

## Methods

### Literature search and study selection

This systematic review was performed according to the recommendations of the PRISMA statement [21] (S1 Checklist).

The PubMed, Cochrane Library, and Web of Science online databases were systematically searched June 1999 to December 2020. The search was performed using the following query: (dengue) and ((shock) or (severe) or (severity) or (dss) or (dhf) or (dengue shock syndrome) or (dengue haemorrhage fever)). Moreover, the references of included studies and relevant reviews were manually retrieved to collect more studies.

Studies that met the following criteria were included: (1) dengue infections were confirmed by laboratory tests; (2) there were SD and DF groups with characteristic data, such as

epidemiological factors, clinical signs, and laboratory parameters; (3) the studies provided original data; (4) the papers were written in English.

Studies meeting the following criteria were excluded: (1) papers with unavailable full texts or data; (2) case reports, reviews, animal studies and in vitro studies; (3) genetic studies; (4) duplicate publications.

All titles and abstracts were first independently reviewed by two authors. The full texts of studies that were potentially eligible to be included were obtained for further reading and scrutiny. Disagreements were resolved by consulting a third author.

## Quality assessment

The Newcastle-Ottawa quality assessment scale (NOS) [22] was used to evaluate the quality of the included studies. Scores were determined by nine metrics: data collection, assignment of the patients, inclusion criteria, exclusion criteria, characteristics of the patient population, interpretation of other characteristics, methodological quality, interpretation of factors and dengue diagnosis. Two authors independently assessed the quality of each original study. Studies were defined as being of low, intermediate, and high quality according to NOS scores of 1–3, 4–6, and 7–9, respectively. The scoring system is available in S1 Table.

## Data extraction

Data were independently extracted by two authors if that were presented at least two studies, and the following information was included: the first author, publication date, country/city of origin, patient recruitment period, age of patients, data type, diagnostic method, criteria for diagnosis, sampling time, quality score, and number of cases. DHF, DSS, and SD were defined collectively as SD in this study. Data that could not be reliably extracted or that overlapped were excluded. When duplication was noted, the largest data set was chosen for the meta-analysis. The information is recorded in S2 Table.

## Statistical analyses

A meta-analysis for predictive factors was carried out using STATA version 12.0 (STATA Corporation, College Station, TX, USA). Heterogeneity was assessed using the Cochran Q test with its corresponding $p$ values and $I^2$ statistic. $I^2$ values of 25%, 50%, and 75% indicated low, moderate, and high levels of heterogeneity, respectively. Heterogeneity was considered statistically significant if the $p$ value was $\leq 0.10$ and $I^2$ was >40% [23, 24]. A random-effect model was used when there was significant heterogeneity; otherwise, a fixed-effect model was used [25]. Dichotomous and continuous variables were analyzed by calculating the pooled odds ratio (OR) and standardized mean difference (SMD), respectively, with 95% confidence intervals (CI) using a fixed or random effect model.

To explore the potential sources of high heterogeneity among the studies, subgroup analyses and meta-regression were performed for sampling time ($\leq 7$ days after onset), the population, data type, criteria for diagnosis, area of origin and study quality, when there were more than ten datasets included [26, 27]. The effect of co-variants was considered significant when $p$ was < 0.05 or the 95% CI did not overlap with the original data.

Publication bias was assessed by Begg's funnel plot and Egger's linear regression test when there were more than ten datasets included [28, 29], and the trim and fill method from Duvall and Tweedie was used by adding studies that appeared to be missing to enhance the symmetry when publication bias was found ($p<0.05$) [30]. The adjusted pooled effect size and 95% CI were computed after adding the potential missing studies. In addition, the sensitivity analysis

was carried out using the leave-one-out method to test whether a potential outlier within the included studies could have influenced the meta-analysis summary effects [31].

Previous studies showed that dengue virus was an important cause of childhood and adult morbidity in Asian and Latin American countries [32] and people with African ancestry were less susceptible to the severe manifestations of dengue infection [33, 34]. Therefore, the subgroups of Asia and America were compared in the Meta-analysis. And sensitivity and subanalysis of co-variables on the summary effect and heterogeneity were performed for factors with more than ten studies included using the one study omitting analyses to test whether a potential outlier within included studies could have influenced the meta-analysis summary effects [31].

## Results

A total of 6848 studies were identified after the initial search of the databases. After the screening of titles and abstracts, 364 potentially relevant papers were retrieved for detailed assessment, and 87 studies with 35,184 DF and 8,173 SD cases were included in the meta-analysis based on the inclusion and exclusion criteria. A total of 34 factors were found to be significantly different between DF and SD, age, diabetes history, secondary infection, seroDENV-2/3, bleeding, vomiting, ascites, pleural effusion, lethargy and petechiae, were positive associated with SD; HCT, ALT, AST, CK, BUN, LDH, IL-10, IL-8, sVCAM-1, and IP-10 were increasing but total protein, albumin and PLT were decreasing in level during SD. The study selection flow diagram is depicted in **Fig 1**.

### Identification of studies

A total of 87 studies published from January 2000 to December 2020 were ultimately included in the study, and 66 (75.9%), 19 (21.8%), 1 (1.1%), and 1 (1.1%) study originated from Asia, the Americas, Europe, and Oceania, respectively. The WHO 1997 [2], WHO 2009 [1], WHO 2011 [35], WHO 1999b [36] and Brazilian guidelines [37] were used for identifying SD in 53 (60.9%), 25 (28.7%), 5 (5.7%), 1 (1.1%) and 3 (3.4%) studies, respectively. The final articles consisted of 53 (60.9%) retrospective, 27 (31.0%) prospective and 7 (8.0%) cross-sectional studies. Based on the NOS scores, 30(34.5%), 55 (63.2%) and 2 (2.3%) studies were of high, intermediate and low quality, respectively (S2 Table). Twenty-two (25.3%) studies reported a population of children, 27 (31.0%) reported adult populations, 31 (35.6%) reported both and 7 (8.0%) did not describe the populations. Fifty-one studies stated the sampling time, of which 23 stated it was less than 7 days after onset when the samples were drawn. The details of the included studies are presented in S3 Table. Twenty-eight factors ($I^2$>40%) were analyzed for sensitivity and 24 factors were heterogeneous in the subgroup meta-analysis except age and gender in population, hepatomegaly, vomit, and pleural effusion in sampling time (S4 Table).

### Systematic analysis and meta-analysis

The data sets for 64 factors were extracted from at least two studies. Thirty-four factors were significantly different between patients with DF and those with SD (Table 1), and 30 factors were not correlated with severity (S5 Table). A total of 21 factors were identified and 9 revealed positive association with SD within 7 days after onset in the timing subgroup analysis (S5 Table).

**Characteristics of the populations.** Age, gender, and diabetes history were identified. After pooling relevant studies, age and diabetes history were positively associated with SD in 46 (SMD = 0.151, 95% CI: 0.027–0.275, $p$ = 0.017) and 9 (OR = 4.418, 95% CI: 2.698–7.232, $p$<0.001) studies with high heterogeneity ($I^2$ = 82.4%, $p$<0.001; $I^2$ = 80.4%, $p$<0.001),

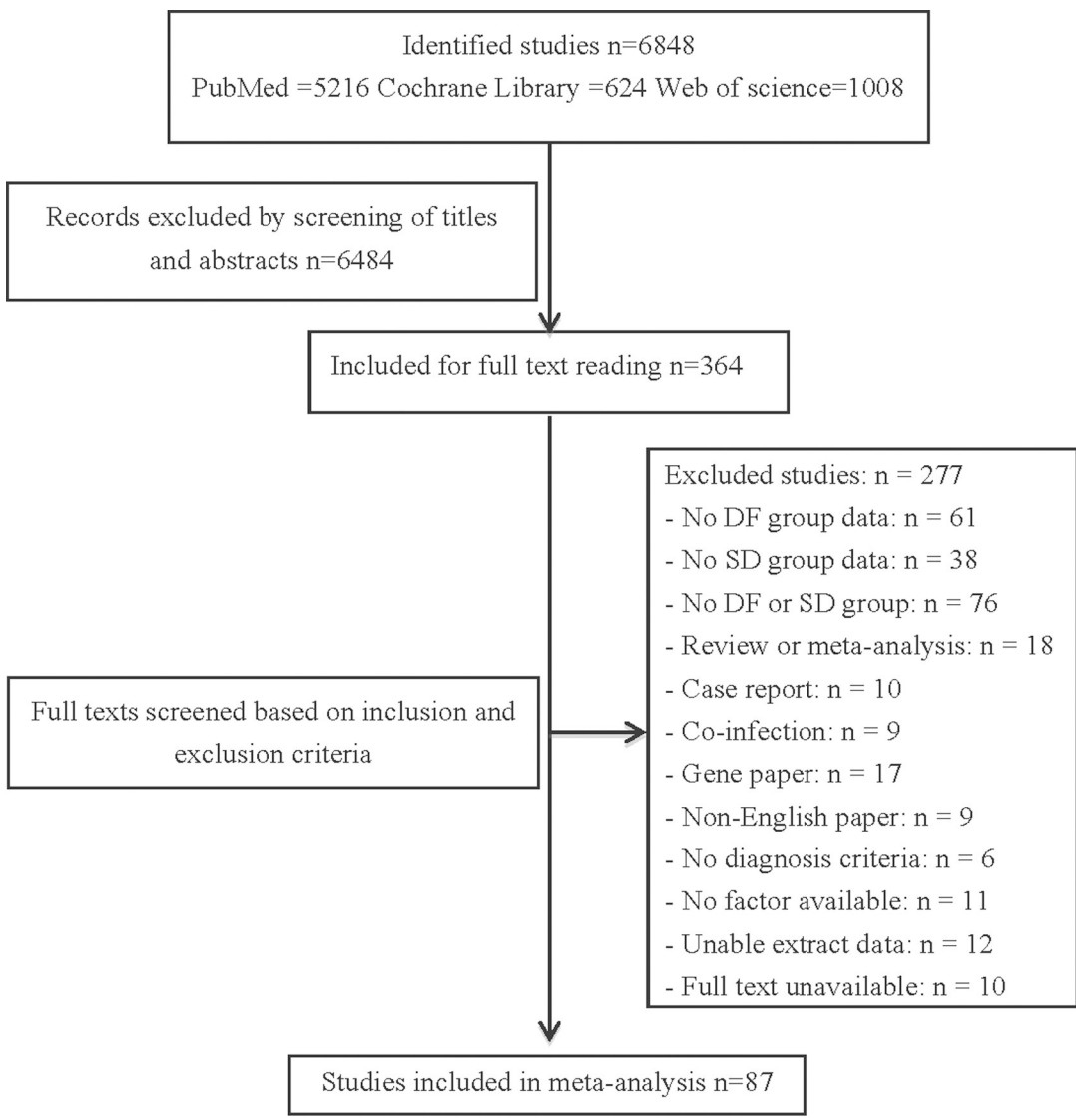

**Fig 1. Flow diagram of the study selection process.**

respectively. Furthermore, meta-regression analysis revealed that the population and sampling time contributed to the heterogeneity of age. However, based on the subgroup analyses, childhood had no correlation with severity in 10 studies (SMD = 0.004, 95% CI: -0.096–0.104, $p = 0.679$) without heterogeneity ($I^2 = 0.0\%$, $p = 0.679$); and age revealed no correlation with severity in 5 studies (SMD = 0.048, 95%CI: -0.192–0.095, $p = 0.510$) with low heterogeneity ($I^2 = 10.5\%$, $p = 0.346$) within 7 days after onset. Additionally, gender did not correlate with SD (S6 Table). Summary effects did not change significantly when the leave-one-out analyses were conducted.

   **Viral characteristics.**   Eighteen studies encompassing 7,659 cases reporting the dengue serotypes together with their severity were obtained, 13 of which originated from Asia and 5 from the Americas. The prevalence rates of DENV-1, DENV-2, DENV-3, and DENV-4 were 39.9%, 29.1%, 19.6% and 11.3%, respectively. A similar seroprevalence distribution in 6,847 cases in Asia was found, with rates of 37.1%, 31.6%, 19.8% and 11.6%, respectively. In contrast,

**Table 1. Positive factors associated with SD.**

| Factors | Studies included | Sample size (SD/DF) | Model | Association with SD | | Test of Heterogeneity | | Publication bias p-value | |
|---|---|---|---|---|---|---|---|---|---|
| | | | | OR/SMD (95% CI) | $p$-value | $I^2$ (%) | $p$-value | Egger's | Begg's |
| Age | 46 | 2655/11000 | Random | SMD = 0.151 (0.027–0.275) | 0.017 | 82.4 | <0.001 | 0.763 | 0.507 |
| Diabetes | 9 | 1560/4844 | Random | OR = 4.418 (2.698–7.232) | <0.001 | 80.4 | <0.001 | - | - |
| Secondary infection | 22 | 3140/21149 | Random | OR = 2.693 (2.083–3.481) | <0.001 | 67.3 | <0.001 | 0.358 | 0.463 |
| SeroDENV-1 | 15 | 2691/4462 | Random | OR = 0.709 (0.504–0.997) | 0.048 | 73.4 | <0.001 | 0.023 | 0.921 |
| SeroDENV-2 | 17 | 2690/4814 | Random | OR = 1.843 (1.269–2.678) | 0.001 | 76.2 | <0.001 | 0.239 | 0.753 |
| SeroDENV-3 | 16 | 2597/4424 | Random | OR = 0.694 (0.492–0.979) | 0.037 | 54.9 | 0.004 | 0.332 | 0.113 |
| Day of illness | 21 | 1218/3220 | Random | SMD = 0.614 (0.346–0.882) | <0.001 | 91.0 | <0.001 | 0.084 | 0.097 |
| Lethargy | 8 | 812/29412 | Random | OR = 2.563 (1.517–4.329) | <0.001 | 83.6 | <0.001 | - | - |
| Vomit | 26 | 2235/9417 | Random | OR = 1.533 (1.203–1.953) | 0.001 | 76.2 | <0.001 | 0.107 | 0.107 |
| Persistent vomiting | 3 | 65/813 | Fixed | OR = 5.569 (3.041–10.200) | <0.001 | 0.0 | 0.835 | - | - |
| Diarrhea | 16 | 1123/3750 | Fixed | OR = 1.245 (1.008–1.537) | 0.042 | 15.8 | 0.273 | 0.383 | 0.096 |
| Abdominal pain | 33 | 2774/27727 | Random | OR = 1.850 (1.466–2.335) | <0.001 | 77.7 | <0.001 | 0.052 | 0.119 |
| Hepatomegaly | 17 | 1601/20581 | Random | OR = 4.403 (3.016–6.430) | <0.001 | 63.9 | <0.001 | 0.135 | 0.592 |
| Petechiae | 19 | 1148/3529 | Random | OR = 2.508 (1.720–3.655) | <0.001 | 57.2 | 0.001 | 0.101 | 0.093 |
| Bleeding | 32 | 2748/27000 | Random | OR = 6.856 (4.160–11.300) | <0.001 | 89.9 | <0.001 | 0.768 | 0.168 |
| Pleural effusion | 19 | 1751/3666 | Random | OR = 15.836 (6.974–35.967) | <0.001 | 87.3 | <0.001 | 0.002 | 0.529 |
| Ascites | 12 | 1271/2213 | Random | OR = 24.299 (4.337–136.138) | <0.001 | 90.9 | <0.001 | 0.001 | 0.837 |
| Hypotension | 11 | 714/1804 | Random | OR = 3.692 (1.670–8.162) | 0.001 | 69.7 | <0.001 | 0.006 | 0.062 |
| HCT | 27 | 1791/7612 | Random | SMD = 0.327 (0.109–0.546) | 0.003 | 91.8 | <0.001 | 0.699 | 0.404 |
| High HCT * | 7 | 607/18180 | Random | OR = 12.389 (6.091–25.199) | <0.001 | 81.2 | <0.001 | - | - |
| PLT | 38 | 2586/26476 | Random | SMD = -1.070 (-1.293- -0.848) | <0.001 | 94.3 | <0.001 | 0.01 | 0.012 |
| Low PLT* | 12 | 728/1238 | Random | OR = 8.146 (3.374–19.665) | <0.001 | 84.8 | <0.001 | 0.063 | 0.161 |

(*Continued*)

**Table 1.** (Continued)

| Factors | Studies included | Sample size (SD/DF) | Model | Association with SD | | Test of Heterogeneity | | Publication bias p-value | |
|---|---|---|---|---|---|---|---|---|---|
| | | | | OR/SMD (95% CI) | p-value | $I^2$ (%) | p-value | Egger's | Begg's |
| ALT | 30 | 1920/23694 | Random | SMD = 1.007 (0.386–1.627) | 0.001 | 99.1 | <0.001 | 0.245 | 0.003 |
| High ALT* | 8 | 528/1069 | Random | OR = 4.030 (2.408–6.747) | <0.001 | 66.1 | 0.004 | - | - |
| AST | 29 | 1888/25527 | Random | SMD = 1.278 (0.640–1.916) | <0.001 | 99.2 | <0.001 | 0.338 | 0.011 |
| High AST | 4 | 129/366 | Fixed | OR = 4.053 (2.255–7.287) | <0.001 | 0.0 | 0.774 | - | - |
| CK | 4 | 65/404 | Random | SMD = 2.647 (1.117–4.177) | 0.001 | 94.9 | <0.001 | - | - |
| ALB | 13 | 972/21740 | Random | SMD = -0.767 (-0.989- -0.544) | <0.001 | 86.8 | <0.001 | 0.006 | 0.008 |
| Low ALB* | 2 | 54/161 | Fixed | OR = 20.601 (4.441–95.562) | <0.001 | 12.6 | 0.285 | - | - |
| TP | 5 | 484/3390 | Random | SMD = -0.271 (-0.449 - -0.093) | 0.003 | 60.4 | 0.039 | - | - |
| Low TP* | 2 | 28/72 | Fixed | OR = 10.993 (2.949–40.978) | <0.001 | 0.0 | 0.443 | - | - |
| Proteinuria | 2 | 528/1098 | Random | OR = 3.681 (2.038–6.649) | <0.001 | 80.1 | 0.025 | - | - |
| BUN | 4 | 361/2966 | Random | SMD = 1.301 (0.330–2.273) | 0.009 | 97.8 | 0.025 | - | - |
| LDH | 5 | 111/469 | Random | SMD = 1.873 (0.494–3.253) | 0.008 | 96.6 | 0.025 | - | - |
| PT | 6 | 242/2611 | Random | SMD = 0.781 (0.219–1.343) | 0.006 | 90.6 | <0.001 | - | - |
| APTT | 6 | 229/2089 | Random | SMD = 0.529 (0.046–1.013) | 0.032 | 80.6 | <0.001 | - | - |
| IL-10 | 6 | 289/425 | Random | SMD = 0.868 (0.197–1.539) | 0.011 | 92.7 | <0.001 | - | - |
| IL-8 | 3 | 127/151 | Random | SMD = 3.337 (1.059–5.615) | 0.004 | 97.3 | <0.001 | - | - |
| sVCAM-1 | 2 | 37/70 | Fixed | SMD = 1.297 (0.856–1.737) | <0.001 | 0.0 | 0.441 | - | - |
| IP-10 | 2 | 92/86 | Random | SMD = 0.531 (0.059–1.004) | 0.027 | 52.9 | 0.145 | - | - |

Pooled odds ratios (OR) or standardized mean difference (SMD) with corresponding 95% confidence intervals (95% CI) of the published results were calculated when the factor was included in more than one study.

* Dichotomous variables

in 812 cases from the Americas, the seroprevalence rates were 63.9%, 8.3%, 18.2% and 9.6%, respectively.

After pooling 17 studies, DENV-2 was positively associated with SD (OR = 1.843, 95% CI: 1.269–2.678, p = 0.001), whereas DENV-1 and DENV-3 had a negative association in 15 (OR = 0.709, 95% CI: 0.504–0.997, p = 0.048) and 16(OR = 0.694, 95% CI: 0.492–0.979,

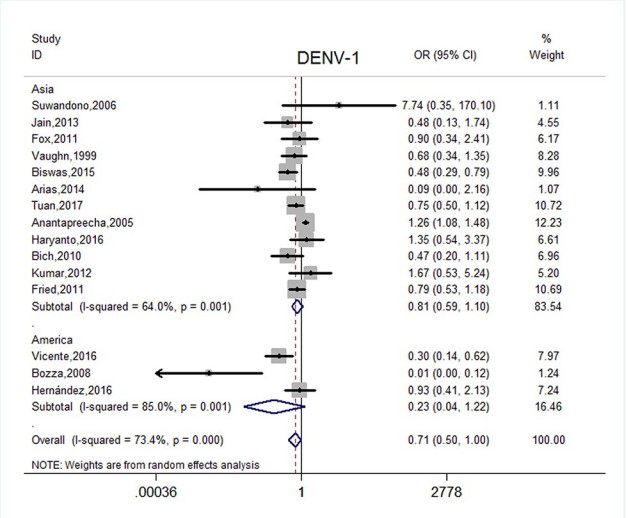

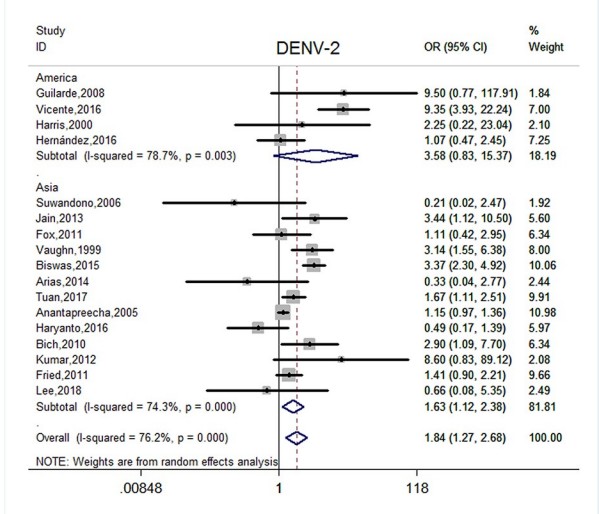

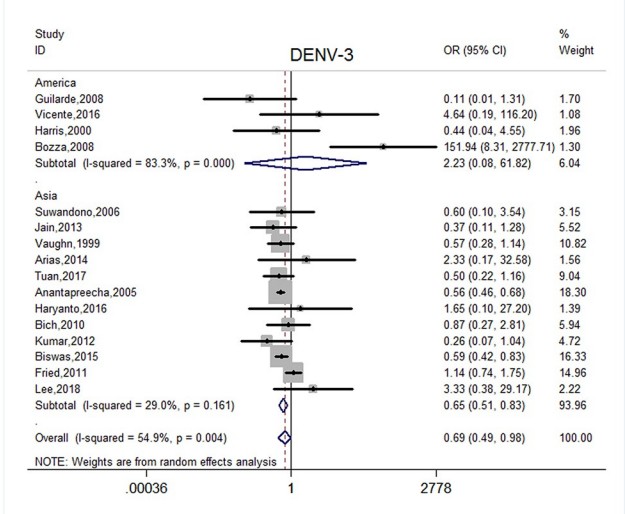

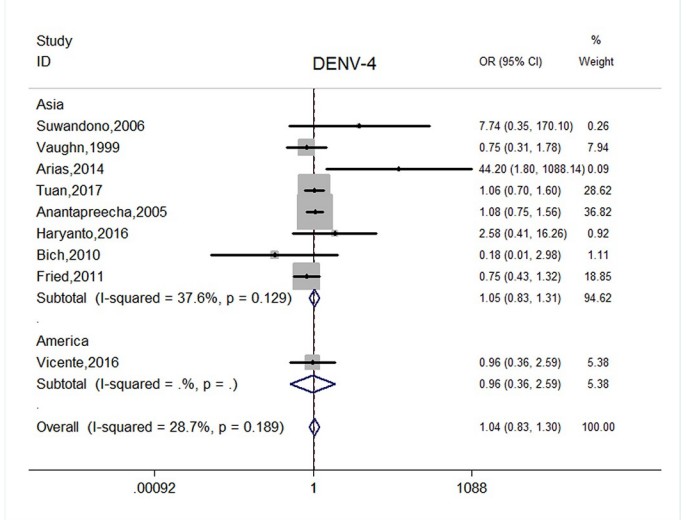

**Fig 2. Forest plot of the subgroup analysis of the area of origin for serotypes of DENV (DF vs SD, OR = odds ratio).** A: DENV-1; B: DENV-2; C: DENV-3; D: DENV-4.

*p* = 0.037) studies, respectively. However, in the subgroup analysis of epidemic areas, DENV-1 revealed an inconsistent result with SD in Asia (OR = 0.810, 95% CI: 0.594–1.104, *p* = 0.182) and in the Americas (OR = 0.230, 95% CI: 0.044–1.215, *p* = 0.084); DENV-3 revealed a similar result with SD (OR = 0.650, 95% CI: 0.511–0.828, *p*<0.001) in Asia but opposite in the Americas (OR = 2.226, 95% CI: 0.080–61.821, *p* = 0.637). DENV-4 showed no significant difference between the two groups in 9 studies. The details are presented in **Fig 2**. In addition, the pooled odds ratio of secondary infection in 22 studies revealed a positive association with SD (OR = 2.693, 95% CI: 2.083–3.481, *p*<0.001). Also, it revealed a consistent association with SD in 4 studies (OR = 2.448, 95% CI: 0.955–6.277, *p* = 0.062) within 7 days after onset. Excluding individual studies did not change the summary effects significantly.

**Clinical manifestations.**   Days of illness was observed to be much longer in SD (SMD = 0.614, 95% CI: 0.346–0.882, *p* = 0.000) after pooling 21 studies. Lethargy/dizziness had a positive association with SD (OR = 2.563, 95% CI: 1.517–4.329, *p*<0.001) after pooling

data from 8 studies. Vomiting and abdominal pain were observed to be risk factors for SD in 26 (OR = 1.533, 95% CI: 1.203–1.953, $p$ = 0.001) and 33 (OR = 1.850, 95% CI: 1.466–2.335, $p$<0.001) studies, respectively, with high heterogeneity. In particularly, persistent vomiting as one of the WS was referred to in 3 studies and had a strong positive pooled effect (OR = 5.569, 95% CI: 3.041–10.000, $p$<0.001). Diarrhea was also associated with SD in 16 studies (OR = 1.245, 95% CI: 1.008–1.537, $p$ = 0.042) with low heterogeneity ($I^2$ = 15.8%, $p$ = 0.273). Additionally, hepatomegaly was highly correlated with SD in 17 studies (OR = 4.403, 95% CI: 3.016–6.430, $p$<0.001). Hepatomegaly revealed a similar association with SD in 2 (OR = 9.264, 95% CI: 7.034–12.201, $p$<0.001) studies with low heterogeneity ($I^2$ = 0.0%, $p$ = 0.402) within 7 days after onset. The high heterogeneity and summary effect did not change significantly when subgroup analyses of other covariables and leave-one-out analyses were conducted.

**Bleeding signs.**   Skin rash, petechiae, hematemesis, melena, gum bleeding, epistaxis, and the tourniquet test were identified as bleeding signs in this study. Severe bleeding, including hematemesis, melena, gum bleeding, and epistaxis, had a strong association with SD (OR = 6.856, 95% CI: 4.160–11.300, $p$ = 0.000) after pooling data from 32 studies. It showed a positive association with SD (OR = 8.106, 95% CI: 3.094–21.241, $p$<0.001) as well when the sampling time subgroup analysis was performed in 7 studies. Additionally, petechiae had a positive association (OR = 2.508, 95% CI: 1.720–3.655, $p$ = 0.000) after pooling data from 19 studies with moderate heterogeneity ($I^2$ = 57.2%, $p$ = 0.001).

**Plasma leakage.**   Pleural effusion and ascites had a strong association with SD after pooling data from 19 (OR = 15.838, 95% CI: 6.974–35.967, $p$<0.001) and 12 (OR = 24.299, 95% CI: 4.337–136.138, $p$<0.001) studies, respectively. However, there was publication bias in favor of positive studies according to Egger's test ($p$<0.05) for both. Using the trim and fill method from Duval and Tweedie, no studies was added for ascites, and the positive association remained after 7 missing studies were added for pleural effusion (original OR = 2.731, 95% CI: 1.939–3.521, $p$ = 0.000; adjusted OR = 1.823, 95% CI: 1.114–2.533, $p$ = 0.000). Both revealed a stronger association with SD within 7 days after onset in 2 studies (OR = 87.143, 95% CI: 10.962–693.405, $p$<0.001; OR = 83.578, 95% CI: 3.786–1844.938, $p$ = 0.005). The details are described in **Fig 3**. Additionally, hypotension was observed to be a risk factor for SD (OR = 3.692, 95% CI: 1.670–8.162, $p$ = 0.001) in 11 studies, and publication bias was found. Using the trim and fill method from Duval and Tweedie, 4 missing studies were added, and the association remained unchanged (original OR = 0.672, 95% CI: 0.318–1.025, $p$<0.001; adjusted OR = 0.452, 95% CI: 0.110–0.793, $p$ = 0.010). The high level of heterogeneity was not reduced, and the summary effect changed significantly when the leave-one-out analyses and subgroup analyses of covariables were conducted.

**Blood cell counts.**   Among the markers investigated, a decrease in the platelet count was observed in 38 studies and was revealed to be a risk factor for SD (SMD = -1.070, 95% CI: -1.293–0.848, $p$<0.001). However, publication bias was observed ($p$<0.05). After 7 missing studies were added, the association was stronger (original SMD = -1.070, 95% CI: -1.293–0.848, $p$<0.001; adjusted SMD = -1.384, 95% CI: -1.665–1.102, $p$<0.001). Furthermore, another twelve dichotomous datasets were pooled and revealed that thrombocytopenia was strongly associated with SD (OR = 8.146, 95% CI: 3.374–19.665, $p$<0.001). Additionally, the quantitative analysis showed that hematocrit (HCT) was positively associated with SD (SMD = 0.327, 95% CI: 0.109–0.546, $p$ = 0.003) in 27 studies, and there were 7 dichotomous datasets with elevated HCT levels, which was strongly associated with SD (OR = 12.389, 95% CI: 6.091–25.199, $p$<0.001). Moreover, sampling time (≤7 days after onset) subgroup analysis was performed and platelet count, thrombocytopenia and HCT were also observed to be risk factors of SD in 10 (SMD = -1.452, 95% CI: -1.872- -1.031, $p$<0.001), 3 (OR = 48.931, 95% CI: 1.873–1278.431, $p$<0.001), 7 (SMD = 0.706, 95% CI: 0.122–1.291, $p$ = 0.018) studies, respectively.

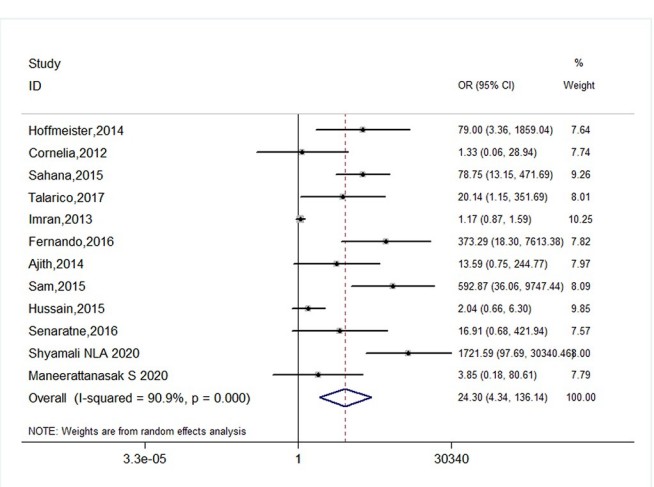
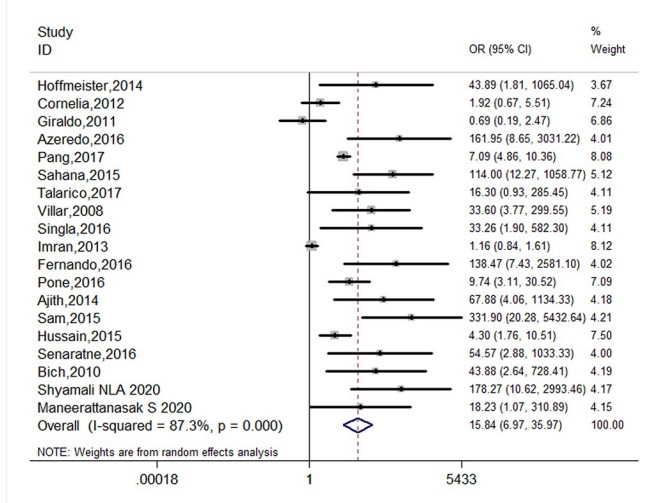

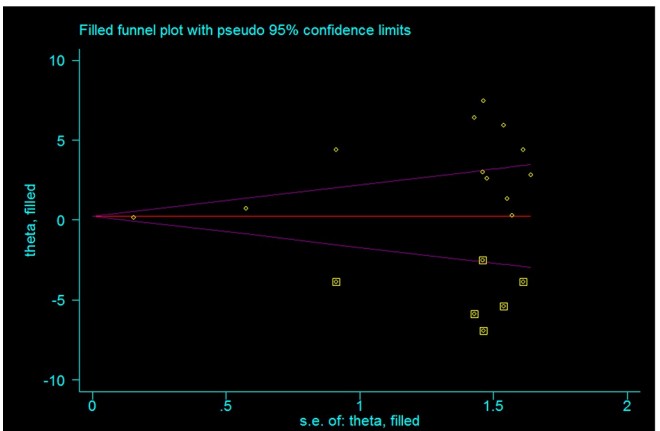
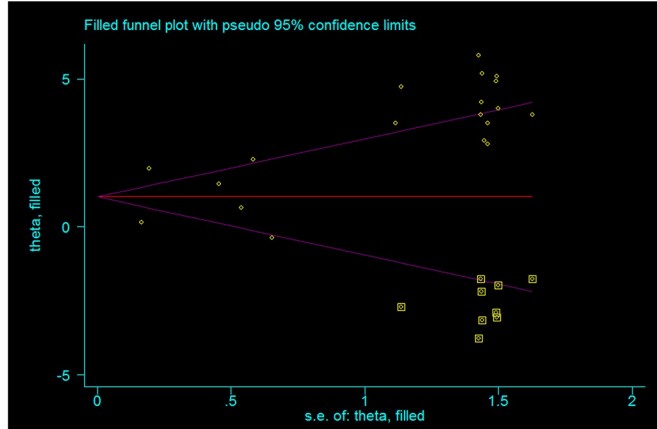

**Fig 3. Association between plasma leakage and SDD.** (A, B) Forest plot for pleural effusion, ascites respectively DF vs SDD, OR: odds ratio. (C, D) Funnel Plots for pleural effusion, ascites respectively (Trim and Fill) DF vs SDD, SE: standardized error.

**Hepatic and renal manifestations, lipids.** Within the list of serum markers, the levels of alanine aminotransferase (ALT) and aspartate aminotransferase (AST) were significantly higher in patients with SD than in those with DF (SMD = 1.007, 95% CI: 0.386–1.627, $p = 0.001$; SMD = 1.278, 95% CI: 0.640–1.916, $p < 0.001$); Also, AST revealed a stronger association with SD in 7 studies (SMD = 1.712, 95% CI: 0.276–3.148, $p = 0.019$) within 7 days after onset. Additionally, the summary effect of elevated ALT and AST also showed a stronger association with SD after pooling 8 (OR = 4.030, 95% CI: 2.408–6.747, $p < 0.001$) and 4 (OR = 4.053, 95% CI: 2.255–7.287, $p < 0.001$) studies, respectively. However, publication bias was observed on Begg's test for ALT ($p < 0.05$) and AST ($p < 0.05$); no study was added using the trim and fill method. Albumin (ALB) and total protein (TP) levels were significantly lower in patients with SD than in those with DF after pooling 13 (SMD = -0.767, 95% CI: -0.989–0.544, $p < 0.001$) and 5 (SMD = -0.271, 95% CI: -0.449–0.093, $p = 0.003$) studies, respectively. Meanwhile, hypoproteinemia, hypoalbuminemia, proteinuria, and increased levels of creatine kinase (CK), lactate dehydrogenase (LDH) and blood urea nitrogen (BUN) were positively associated with SD (Table 1).

**Coagulation tests.** Prolonged prothrombin time (PT) and activated partial thromboplastin time (APTT) were found to be significantly associated with SD after pooling 6 studies (SMD = 0.781, 95% CI: 0.219–1.343, $p$ = 0.006; SMD = 0.529, 95% CI: 0.046–1.013, $p$ = 0.032). However, the summary effect of prolonged PT, prolonged APTT, and elevated D-dimer levels had a negative association with SD in two dichotomous datasets (S5 Table).

**Cytokines and chemokines.** Various detection methods and descriptions of the results were observed in the original literature. A wide blood sampling window was observed, ranging from the acute phase to the convalescence phase. Studies with mean differences in cytokines and chemokines available were selected for the current study. Overall, eleven cytokines and chemokines were identified after pooling the relevant studies. Levels of IL-10, IL-8, sVCAM-1, and IP-10 were positively associated with SD in 6, 3, 2 and 2 studies, respectively (Table 1). Furthermore, subgroup analyses of sampling time showed inconsistent results for IFN-γ. The level was significantly higher in 4 studies (SMD = 0.184, 95% CI: 0.023–0.344, $p$ = 0.025) without heterogeneity ($I^2$ = 0.0%, $p$ = 0.480) when sampled early in the disease course ($\leq$7 days of onset). Additionally, most cytokines (except sVCAM-1 and IP-10) had discordant results in the included individual studies.

## Discussion

It is critically important to identify the predictive factors for SD, as the early diagnosis and treatment of SD could reduce mortality and decrease hospitalization durations and costs. The pathogenesis of SD is multifactorial and is not yet well understood. Antibody-dependent enhancement (ADE) due to non-neutralizing cross-reactive antibodies may play a vital role in the mechanism, especially in secondary infection cases [38, 39]. Zhang H, et al [40] conducted a meta-analysis that provided the evidence for the classifications of severe dengue disease according to the new WHO guideline 2009 based on the literature between 2000 and 2012. However, compared with symptoms and signs, virus serotype and plasma biomarkers results were obtained more objectively. In this study, 34 factors including clinical manifestations, virus serotypes, medical history, and plasma biomarkers, were found to be significantly different between DF and SD. Since the critical phase of dengue is usually on days 3–7 of illness [1], subgroups analysis for sampling time ($\leq$7 days after onset) were performed in this study. Nine factors revealed association with SD within 7 days after onset and could be predictors for SD.

### Clinical manifestations

It was further confirmed that SD was associated with secondary infections in the current study, indicating that DF patients with secondary infection had a 2.69 times higher risk of SD than those with only DF. The WS were further confirmed in the current meta-analysis; hepatomegaly, bleeding, pleural effusion, ascites, and persistent vomiting were associated with 4.4, 6.9, 15.8, 24.3, and 5.6 times the risk of SD, respectively, which were consistent with previous study [40]. Thus, patients with WS should be treated appropriately and in a timely manner to prevent the development of SD. Moreover, lethargy and hypotension had a positive association with SD, which meant that these manifestations are also predictors of SD. However, significant heterogeneity was observed among studies regarding these clinical manifestations. The heterogeneity might be due to inherent differences in populations.

### Viral and host factors

This finding also showed a clear difference in the associations of dengue serotypes with the percentage of severe cases. Although DENV-1 accounted for the highest percentage of dengue infections, there was a lower risk of SD on overall, whereas, no statistically significant

difference was revealed between patients with DF and those with SD in the Asia and in the Americas, respectively. DENV-2 was a risk factor for SD, even though it had the lowest sero-prevalence in the Americas. However, DENV-3 had an inconsistent association with SD, with a negative association in Asia (OR = 0.669, $p$ = 0.021) and no association in the Americas. However, it is premature to draw firm conclusions. The serotypes of DENV were not always reported in the included studies. Only 20.7% of the studies included provided serotype data, and many did not separate primary from secondary dengue infections caused by each dengue serotype. Meanwhile, discordant results have been observed in other studies. DENV-1 is seldom involved in severe cases in Brazil [41], whereas it was associated with DHF and SD in Singapore [42] DENV-4 was found to be strongly associated with DSS in Brazil [41] and individuals infected with DENV-4 had a higher prevalence of respiratory and cutaneous manifestations in South America [43] Rico-Hesse et al. proposed the term virulent genotypes and revealed an association between two distinct genotypes of DENV-2 and the appearance of DHF in the Americas [44]. Alternatively, it had been reported that the genetic changes in DENV-3 were associated with the increasing severe dengue epidemics in Sri Lanka [45]. Thus, the serotype of DENV can contribute to SD differently based on other factors, and the sero-prevalence [46] and changes in the viral genotype [47] during epidemics might be potential factors affecting the development of SD; this needs further investigation.

In this study, there was no association between age and SD in children, but increasing age results in a higher risk of progressing into SD among adults after pooling 46 studies, agreeing with previous studies[15, 40, 48]. However, consistent conclusions were observed in different populations. For example, pregnant women are 3.4 times more likely to develop SD than non-pregnant women [49]; as high a proportion as 80% of infants hospitalized with dengue developed DHF/DSS [50]. Furthermore, individuals with a history of diabetes had a 4.42 times higher risk of SD than those without a history of diabetes. The reason might be that diabetes could result in immune and endothelial dysfunction [51, 52].

## Plasma biomarkers

Some evidence also indicated that the incidence of low platelet counts, plasma leakage, shock and hemorrhagic manifestations were significantly different in infants compared with older children, and bleeding signs, including rash, petechiae and obvious bleeding were observed approximately 2 times more often in adults than children. [53, 54]. An increase in HCT concurrent with a rapid decrease in platelet count was defined as one of the WS by the WHO [1]. In the current study, thrombocytopenia, an increase in HCT and a decrease in the platelet count were associated with SD, so did in subgroup analysis for sampling time (≤7 days after onset). The leukocyte count is frequently used to evaluate suspected bacterial infections. It had been indicated to be a good marker for differentiating between bacterial versus viral infections in a prospective study [55]. In the early febrile phase, a decreasing white blood cell count makes the diagnosis of dengue very likely [1]. However, the counts of the total population and subpopulations of white blood cells were not different between patients with DF and those with SD. As the whole blood counts were dynamic throughout the pathogenic process, a subgroup analysis of sampling time was conducted, but the pooled effect showed no significant difference. Additionally, some studies revealed that atypical lymphocyte count, immature platelet fraction and triple positivity for NS1, Ig M and Ig G would be predictive for SD [56–58], although them couldn't be included in this meta-analysis. Further studies should be performed to identify in the future.

Liver damage is a well-established characteristic of dengue patients, particularly in severe cases[1, 59], and ALT or AST≥1000 IU/L is a diagnostic criterion for SD [1]; these facts

highlight that the liver is involved in the pathogenesis of dengue infection. In this meta-analysis, elevated ALT levels, elevated AST levels and hypoalbuminemia were positively associated with SD. Furthermore, the levels of LD, CK and BUN were increased in patients with SD compared with patients with DF. Unfortunately, there were not enough clinical data available to determine the cutoff values of the indicators. Thus, more clinical studies with defined cutoff values are needed to address these biomarkers in the future.

It is well known that cytokines and chemokines play important roles in the pathogeny of dengue infection but inconsistent association between DF and SD was observed in the literature because of the heterogeneity [13, 60, 61]. In the current study, the pooled results of IL-8, IL-10, sVCAM-1 and IP10 were positively associated with severity. However, these findings should be interpreted cautiously because conflicting results were observed among studies. One of the major hindrances is the inconsistent results in the literature caused by heterogeneity. Large variations were observed among studies with various sampling times. Regarding IFN-γ, a significant positive association with SD was revealed in the acute phase after removing studies with different sampling times. Additionally, the level of IFN-γ was significantly higher in patients with SD than in those with DF, but opposing results were observed during the defervescence and convalescent stages [62, 63]. As the levels of cytokines/chemokines are dynamic during the process of infection, they display differences in the timing of their peak responses [64, 65]. Thus, different factors should be measured during the appropriate phases. Furthermore, there were significant differences in the levels of IL-8 and VEGFR2 between serum and plasma samples [64, 65]. These results merit further investigation with better-defined methodologies, full descriptions of the results and transparency of the sampling time and serotypes. These data would be helpful in overcoming the weaknesses of the currently available publications.

## Limitations

There were some limitations in our study. First, there were some markers, such as viremia, nutritional status, and serum levels of C-reactive protein, total cholesterol, and triglycerides, were not analyzed in this study because of insufficient data. Second, the significant heterogeneity was not fully explained by the six covariables investigated. It could have been driven by numerous other factors that were not addressed in this meta-analysis, which shows the need of controls for these factors in order to further confirm the findings in future research. Third, some reasons might conduct biases, such as most reports were retrospective, non-English studies were excluded, samples were processed into plasma or serum and different WHO classification methods were used to assign the disease's severity.

## Conclusion

A list of 34 potential severity markers was investigated in this study; and nine factors, secondary infection, retro orbital pain, hepatomegaly, bleeding, pleural effusion, ascites, increased HCT, and AST, decreased PLT revealed positive relation with SD in early stage (≤7 days after onset). Hence, this study provides information regarding markers that can be used to identify SD in the early stage, facilitating prompt disease management. However, heterogeneity was observed among current studies, which suggests that increased standardization is needed in future clinical reports.

## Supporting information

**S1 Checklist. PRISMA checklist.**
(DOC)

**S1 Table. NOS scoring system for quality assessment.**
(DOC)

**S2 Table. The scores of studies included in this meta-analysis according to NOS.**
(DOCX)

**S3 Table. Characteristics of the studies included in this meta-analysis.**
(DOCX)

**S4 Table. Sensitivity and sub-analysis on the summary effect and heterogeneity.**
(DOC)

**S5 Table. Factors identified by subgroup analysis for sampling time within 7 days after onset of illness.**
(DOC)

**S6 Table. Factors not associated with sever dengue disease.**
(DOC)

**S1 Data.**
(XLSX)

## Acknowledgments

The authors would like to thank Dr. Peihuang Wu for assistance during the preparation of this manuscript.

## Author Contributions

**Conceptualization:** Kangzhuang Yuan, Yongping Lin, Lidong Liu.

**Data curation:** Kangzhuang Yuan, Yuan Chen, Meifeng Zhong.

**Formal analysis:** Kangzhuang Yuan, Lidong Liu.

**Funding acquisition:** Lidong Liu.

**Investigation:** Kangzhuang Yuan, Yuan Chen, Lidong Liu.

**Methodology:** Kangzhuang Yuan, Yuan Chen.

**Supervision:** Yongping Lin, Lidong Liu.

**Validation:** Kangzhuang Yuan, Yuan Chen, Meifeng Zhong, Yongping Lin, Lidong Liu.

**Writing – original draft:** Kangzhuang Yuan, Yongping Lin, Lidong Liu.

**Writing – review & editing:** Kangzhuang Yuan, Yuan Chen, Meifeng Zhong, Yongping Lin, Lidong Liu.

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
