## [Decision Letter · Decision Letter 0]

28 Feb 2022

PONE-D-21-08797

Risk and predictive factors for severe dengue infection: a systematic review and meta-analysis

PLOS ONE

Dear Dr. Liu,

Thank you for submitting your manuscript to PLOS ONE. After careful consideration, we feel that it has merit but does not fully meet PLOS ONE’s publication criteria as it currently stands. Therefore, we invite you to submit a revised version of the manuscript that addresses the points raised during the review process.

We look forward to receiving your revised manuscript.

Kind regards,

Mao-Shui Wang

Academic Editor

PLOS ONE

Journal Requirements:

2. Please revise your tables to replace p-values of "0.000" to "<0.001.

3. Please attach a Supplemental file of the results of the quality assessment for each individual study assessed, reporting the outcome for each individual criteria considered.

7. We note that you have included the phrase “data not shown” in your manuscript. Unfortunately, this does not meet our data sharing requirements. PLOS does not permit references to inaccessible data. We require that authors provide all relevant data within the paper, Supporting Information files, or in an acceptable, public repository. Please add a citation to support this phrase or upload the data that corresponds with these findings to a stable repository (such as Figshare or Dryad) and provide and URLs, DOIs, or accession numbers that may be used to access these data. Or, if the data are not a core part of the research being presented in your study, we ask that you remove the phrase that refers to these data.

8. Please upload a copy of Supporting Information Tables S1-S4 and Prisma checklist which you refer to in your text on page 20.

9. We note that this manuscript is a systematic review or meta-analysis; our author guidelines therefore require that you use PRISMA guidance to help improve reporting quality of this type of study. Please upload copies of the completed PRISMA checklist as Supporting Information with a file name “PRISMA checklist”.

Reviewers' comments:

Reviewer's Responses to Questions

**Comments to the Author**

1. Is the manuscript technically sound, and do the data support the conclusions?

Reviewer #1: Yes

Reviewer #2: Yes

Reviewer #3: Yes

2. Has the statistical analysis been performed appropriately and rigorously? 

Reviewer #1: Yes

Reviewer #2: Yes

Reviewer #3: N/A

3. Have the authors made all data underlying the findings in their manuscript fully available?

Reviewer #1: Yes

Reviewer #2: Yes

Reviewer #3: Yes

4. Is the manuscript presented in an intelligible fashion and written in standard English?

Reviewer #1: No

Reviewer #2: Yes

Reviewer #3: Yes

5. Review Comments to the Author

Reviewer #1: The manuscript “Risk and predictive factors for severe dengue infection: a systematic review and meta-analysis” by Yuan et al., offers a comprehensive review and reanalysis of the data from the manuscripts published on Dengue Fever, over the last 20 years. The focus of the manuscript is detection of the factors that are correlated with SD with the idea of using those factors to be able to better predict which patients with DF will progress to develop severe form of disease. Early detection and treatment of the severe form of Dengue Fever would lead to significant reduction of mortality.

Authors searched PubMed database for specific articles using very focused search words which yielded almost 7000 manuscripts. Through a multi-step filtering process, final number of 87 manuscripts were selected for inclusion in this study.

A very significant contribution that these types of analyses, and indeed this one as well, provide is closer examination of findings from individual studies in the context of much larger pool of data. Authors have used relatively stringent parameters to exclude a number of different correlates, but also provided stronger association for a number of other ones. Overall, as it would be expected, no new factors showing correlation with the SD were found, but it is possible that a significance of some factors has been emphasized.

Overall, as such, the manuscript does have a significant contribution to the field. However, there are few changes that can be made. By far, the most obvious one seems to be factoring the timing for the correlates. In the discussion, authors correctly state that it is critically important to identify the predictive factors for SD early, but nowhere in the manuscript they discuss timing for the correlates they mention. I am not sure what data is available on this, but even the crudest grouping would be helpful (early, late). Is there anything that can be concluded about the timing of warning signs relative to the onset of symptoms?

Besides that, there are few parts of the manuscript, most notably introduction and discussion where manuscript would benefit from more clarity in the way it was written. I think that the readability would be greatly enhanced if authors would consider having a technical writer or an editor review the grammar and the style.

With all this in mind, I would recommend publishing this, especially if the above changes are made.

Reviewer #2: I would like to thank and congratulate all authors for conducting timely important systematic review and meta analysis on "Risk and predictive factors for severe dengue infection: I believe results of this study will be much important mainly in resources limiting developing counties where their health resources are stretched to its maximum due to present pandemic.

I have few minor suggestions.

1. In the discussion section authors can mention novel bio marks such as Atypical lymphocyte count has been identified as a predictor for sever dengue infection.

2. Furthermore immature platelet fraction (IPF) too can be a predictor for SD.

3. Furthermore Triple positivity for nonstructural antigen 1, immunoglobulin M and immunoglobulin G is predictive of severe thrombocytopaenia related to dengue infection.

Above mentioned points can be included in discussion section . These may help on completeness of the study and planning of future reviews.

Reviewer #3: If the statistical reviewer is fine with the overall statistical conclusions, the article would be suitable for publication

The article would benefit from further english language editing and simplifying some of the difficult to understand sentences

6. PLOS authors have the option to publish the peer review history of their article (what does this mean?). If published, this will include your full peer review and any attached files.

Reviewer #1: No

Reviewer #2: **Yes: **Visula Abeysuriya

Reviewer #3: No

---

## [Author Response · Author response to Decision Letter 0]

27 Mar 2022

Journal Requirements:

A: Yes, we amended in the main text.

2. Please revise your tables to replace p-values of "0.000" to "<0.001.

A：Amended in the main text and supplemental files.

3. Please attach a Supplemental file of the results of the quality assessment for each individual study assessed, reporting the outcome for each individual criteria considered.

A：The details of the quality assessment were shown in Table S2 which was uploaded as a supplemental file.

A: We would like to change the statement of financial disclosure and the details were included in resubmission cover letter.

A: All relevant data are within the manuscript and its Supporting Information files.

A: We don’t have any data except that were within the manuscript and its Supporting Information files.

7. We note that you have included the phrase “data not shown” in your manuscript. Unfortunately, this does not meet our data sharing requirements. PLOS does not permit references to inaccessible data. We require that authors provide all relevant data within the paper, Supporting Information files, or in an acceptable, public repository. Please add a citation to support this phrase or upload the data that corresponds with these findings to a stable repository (such as Figshare or Dryad) and provide and URLs, DOIs, or accession numbers that may be used to access these data. Or, if the data are not a core part of the research being presented in your study, we ask that you remove the phrase that refers to these data.

A: Amended in the main text and provided relevant data.

8. Please upload a copy of Supporting Information Tables S1-S4 and Prisma checklist which you refer to in your text on page 20.

A: We had uploaded all supporting information Tables S1-S6, Prisma checklist, and raw data.

9. We note that this manuscript is a systematic review or meta-analysis; our author guidelines therefore require that you use PRISMA guidance to help improve reporting quality of this type of study. Please upload copies of the completed PRISMA checklist as Supporting Information with a file name “PRISMA checklist”.

A: Yes, we did.

A: Yes, we checked all references again and all were correct.

Reviewers' comments:

Reviewer's Responses to Questions

Comments to the Author

1. Is the manuscript technically sound, and do the data support the conclusions?

Reviewer #1: Yes

Reviewer #2: Yes

Reviewer #3: Yes

2. Has the statistical analysis been performed appropriately and rigorously?

Reviewer #1: Yes

Reviewer #2: Yes

Reviewer #3: N/A

3. Have the authors made all data underlying the findings in their manuscript fully available?

Reviewer #1: Yes

Reviewer #2: Yes

Reviewer #3: Yes

4. Is the manuscript presented in an intelligible fashion and written in standard English?

Reviewer #1: No

Reviewer #2: Yes

Reviewer #3: Yes

5. Review Comments to the Author

Reviewer #1: The manuscript “Risk and predictive factors for severe dengue infection: a systematic review and meta-analysis” by Yuan et al., offers a comprehensive review and reanalysis of the data from the manuscripts published on Dengue Fever, over the last 20 years. The focus of the manuscript is detection of the factors that are correlated with SD with the idea of using those factors to be able to better predict which patients with DF will progress to develop severe form of disease. Early detection and treatment of the severe form of Dengue Fever would lead to significant reduction of mortality.

Authors searched PubMed database for specific articles using very focused search words which yielded almost 7000 manuscripts. Through a multi-step filtering process, final number of 87 manuscripts were selected for inclusion in this study.

A very significant contribution that these types of analyses, and indeed this one as well, provide is closer examination of findings from individual studies in the context of much larger pool of data. Authors have used relatively stringent parameters to exclude a number of different correlates, but also provided stronger association for a number of other ones. Overall, as it would be expected, no new factors showing correlation with the SD were found, but it is possible that a significance of some factors has been emphasized.

Overall, as such, the manuscript does have a significant contribution to the field. However, there are few changes that can be made. By far, the most obvious one seems to be factoring the timing for the correlates. In the discussion, authors correctly state that it is critically important to identify the predictive factors for SD early, but nowhere in the manuscript they discuss timing for the correlates they mention. I am not sure what data is available on this, but even the crudest grouping would be helpful (early, late). Is there anything that can be concluded about the timing of warning signs relative to the onset of symptoms?

Besides that, there are few parts of the manuscript, most notably introduction and discussion where manuscript would benefit from more clarity in the way it was written. I think that the readability would be greatly enhanced if authors would consider having a technical writer or an editor review the grammar and the style.

With all this in mind, I would recommend publishing this, especially if the above changes are made.

A: We do agree that it is very important to identify the predictive factors for SD in the early stage of illness. Unfortunately, the sampling time in most studies were not clearly stated or just had an interval. Therefore, we performed a subgroup analysis for sampling time (we define “early stage” for ≤7 days after onset of illness) in the new manuscript we submitted. A total of 18 factors were identified and 9 revealed positive association with SD in the early stage of illness, which were added in the main text and shown in supplement file (Table S5).

The manuscript was reviewed by an English-speaker editor for grammar and style and all modifications were tracked in the main text. 

Reviewer #2: I would like to thank and congratulate all authors for conducting timely important systematic review and meta analysis on “Risk and predictive factors for severe dengue infection: I believe results of this study will be much important mainly in resources limiting developing counties where their health resources are stretched to its maximum due to present pandemic.

I have few minor suggestions.

1. In the discussion section authors can mention novel bio marks such as Atypical lymphocyte count has been identified as a predictor for sever dengue infection.

2. Furthermore immature platelet fraction (IPF) too can be a predictor for SD.

3. Furthermore Triple positivity for nonstructural antigen 1, immunoglobulin M and immunoglobulin G is predictive of severe thrombocytopaenia related to dengue infection.

Above mentioned points can be included in discussion section . These may help on completeness of the study and planning of future reviews.

A: Yes, we got some studies focused on the three factors mentioned by reviewer. However, they could not be included the meta-analysis according to the including criteria of this study. Thus, we added some context in the discussion section referring to some references.

Reviewer #3: If the statistical reviewer is fine with the overall statistical conclusions, the article would be suitable for publication

The article would benefit from further english language editing and simplifying some of the difficult to understand sentences

A: The manuscript was reviewed again by an English-speaker editor for grammar and style and all modifications were tracked in the main text.

6. PLOS authors have the option to publish the peer review history of their article (what does this mean?). If published, this will include your full peer review and any attached files.

Do you want your identity to be public for this peer review? For information about this choice, including consent withdrawal, please see our Privacy Policy.

Reviewer #1: No

Reviewer #2: Yes: VisulaAbeysuriya

Reviewer #3: No

---

## [Decision Letter · Decision Letter 1]

5 Apr 2022

Risk and predictive factors for severe dengue infection: a systematic review and meta-analysis

PONE-D-21-08797R1

Dear Dr. Liu,

We’re pleased to inform you that your manuscript has been judged scientifically suitable for publication and will be formally accepted for publication once it meets all outstanding technical requirements.

Kind regards,

Mao-Shui Wang

Academic Editor

PLOS ONE

Additional Editor Comments (optional):

Reviewers' comments:

Reviewer's Responses to Questions

**Comments to the Author**

1. If the authors have adequately addressed your comments raised in a previous round of review and you feel that this manuscript is now acceptable for publication, you may indicate that here to bypass the “Comments to the Author” section, enter your conflict of interest statement in the “Confidential to Editor” section, and submit your "Accept" recommendation.

Reviewer #2: All comments have been addressed

2. Is the manuscript technically sound, and do the data support the conclusions?

Reviewer #2: Yes

3. Has the statistical analysis been performed appropriately and rigorously? 

Reviewer #2: Yes

4. Have the authors made all data underlying the findings in their manuscript fully available?

Reviewer #2: Yes

5. Is the manuscript presented in an intelligible fashion and written in standard English?

Reviewer #2: Yes

6. Review Comments to the Author

Reviewer #2: I would like to congratulate all authors for their commitment on compiling this very useful review. Authors have addressed all the reviewer comments comprehensively. This systematic review and meta analysis on Sever dengue and its early predictably will be benefited by all the counties which encounter dengue infection thought the world.

7. PLOS authors have the option to publish the peer review history of their article (what does this mean?). If published, this will include your full peer review and any attached files.

Reviewer #2: No

---

## [Editor Report · Acceptance letter]

7 Apr 2022

PONE-D-21-08797R1 

Risk and predictive factors for severe dengue infection: a systematic review and meta-analysis 

Dear Dr. Liu:

I'm pleased to inform you that your manuscript has been deemed suitable for publication in PLOS ONE. Congratulations! Your manuscript is now with our production department. 

Kind regards, 

on behalf of

Dr. Mao-Shui Wang 

Academic Editor

PLOS ONE